# Ipsilateral Lower-to-Upper Limb Cross-Transfer Effect on Muscle Strength, Mechanical Power, and Lean Tissue Mass after Accentuated Eccentric Loading

**DOI:** 10.3390/medicina57050445

**Published:** 2021-05-04

**Authors:** Hashish R. Magdi, Sergio Maroto-Izquierdo, José Antonio de Paz

**Affiliations:** 1Institute of Biomedicine, (IBIOMED), University of León, 24071 León, Spain; magdigchashish@hotmail.com (H.R.M.); japazf@unileon.es (J.A.d.P.); 2Department of Health Sciences, European University Miguel de Cervantes, 47012 Valladolid, Spain

**Keywords:** cross-education, resistance training, eccentric overload, unilateral training, ipsilateral transfer

## Abstract

*Background and Objectives:* To investigate the effects of unilateral accentuated eccentric loading (AEL) on changes in lean mass and function of leg trained (TL) and ipsilateral non-trained arm (NTA) in young men and women. *Materials and Methods:* In a prospective trial, 69 Physically active university students (20.2 ± 2.2 years) were randomly placed into a training group (*n* = 46; 27 men, 19 women) or a control group without training (*n* = 23; 13 men, 10 women). Participants in the training group performed unilateral AEL in the leg press exercise of the dominant leg twice a week for 10 weeks. An electric motor device-generated isotonic resistance at different intensities for both concentric (30% of 1-RM) and eccentric contractions (105% of 1-RM). Changes in thigh and arm lean tissue mass, unilateral leg press and unilateral elbow flexion maximal concentric (1-RM) and isometric strength (MVIC), and unilateral muscle power at 40, 60, and 80% 1-RM for both leg press and elbow flexion exercises before and after intervention were compared between groups, between sexes and between TL and NTA. *Results:* Both men and women in the training group showed increases (*p* < 0.05) in lean tissue mass, 1-RM, MVIC, and muscle power for TL. In NTA, 1-RM, MVIC, and muscle power increased without significant differences between sexes, but neither in men nor women changes in lean tissue mass were observed. In addition, men showed greater changes in TL, but changes in NTA were similar between sexes. No gains in any variable were found for the control group. *Conclusions:* AEL protocol produced similar neuromuscular changes in TL and ipsilateral NTA, which suggests that strong ipsilateral lower-to-upper limb cross-transfer effects were induced by the eccentric-overload training. However, early ipsilateral increases in muscle force and power were not associated with lean mass gains. Both men and women experienced similar changes in NTA; however, men showed greater changes in TL.

## 1. Introduction

The effects of unilateral strength training on the untrained contralateral limb have been widely studied [1,2,3]. This phenomenon is called the cross-education effect [4]. Numerous studies have shown that unilateral resistance training (RT) improves not only the ipsilateral limb muscle strength but also the strength of the untrained contralateral homologous limb muscle [5]. It has been shown that the mean magnitude of the cross-education effect on muscle strength is 21%, with a greater effect (27%) in lower limb muscles when compared with upper limb muscles (13%) [1,2,3]. Although the mechanisms underpinning this effect have not been fully elucidated [6], neurological and muscle mechanisms have been proposed [1]: (a) neural mechanism, based on the theory of producing neuroplasticity in both cerebral cortexes (cross-activation theory), development of new motor engrams due to new motor learning (bilateral access theory), and movement visualization reduces intracortical inhibition of the ipsilateral cerebral cortex (mirror training theory) [7]; (b) muscle mechanisms, based on the circulating anabolic/anticatabolic hormonal responses related to protein synthesis regulated by the Akt-mTOR pathway [8].

Several studies have shown that eccentric training accentuates the cross-education effect [9]. Hortobagyi et al. [10], Coratella et al. [11], and Lepley and Palmieri-Smith [9] reported that eccentric knee-extension isokinetic training had a cross-education effect on eccentric torque of 23, 11, and 46%, respectively, in the untrained leg. Weir et al. [12] also showed that eccentric isotonic RT led to significant gains in the isometric strength of the contralateral limb by 12%. Accordingly, unilateral eccentric RT protocols have shown gains in strength between 47 and 55% [9,13] in the non-trained limb. However, it has recently been shown that eccentrically accentuated training could induce greater effects by providing an optimal stimulus for both concentric and eccentric actions and providing eccentric contractions at high speed [14]. Therefore, the use of isotonic RT with eccentric-overload can be an alternative to facilitate its application and increase its potential neurophysiological effects.

According to the specialized literature, it is widely known that cross-education is specific to the homologous muscle group and the magnitude of the contralateral gains largely depends on those obtained ipsilaterally [2,15]. However, the extent of cross-education effects on heterologous muscles has not been fully elucidated yet. Throughout literature, several authors have reported an enhancement in ipsilateral elbow flexors strength when an upper-body coadjutant training was added to a lower-body RT program [16,17,18]. Lately, Ben Othman et al. [19], after an 8-week training program (24 sessions) involving the unilateral leg press exercise, reported similar effects on strength in both contralateral non-trained homologous muscles and heterologous (elbow flexors) muscles in youth participants. In addition, the greater magnitudes of changes in the contralateral non-trained leg and non-trained arm were shown after training with higher intensities compared to lower intensities [19]. This lower-to-upper cross-body phenomenon could mean that neural changes induced by RT not only generate changes in the strength of the implicated musculature but also in other untrained areas of the body. It would provide worth for the use of the unilateral accentuated eccentric loading (AEL) as a strategy to achieve optimization of the upper limb musculature for both sports and rehabilitation purposes without the need for its training.

However, given the need for more evidence of the extent of lower-to-upper (i.e., ipsilateral heterologous muscles) cross-body transfer effects and the witnessed superiority of eccentric training to promote neural changes that lead to greater cross-education effects on strength. We intended a study to analyze and compare the effects of unilateral lower body AEL on the trained leg and ipsilateral non-trained arm strength-related variables and lean tissue mass in active young adults. With the hypothesis that eccentric-overload training performed by the lower limb may lead to significant gains in neuromuscular variables, which would have an important impact in clinical and rehabilitation settings, specifically with patients who have an immobilized limb or neurological disorders. Thus, this study aimed to analyze the effects of a 10-week unilateral lower limb AEL program on the ipsilateral upper limb strength-related variables and lean tissue mass in young physically active men and women.

## 2. Materials and Methods

### 2.1. General Design

Experimental participants accomplished 20 sessions of eccentrically accentuated unilateral leg press exercise over 10 weeks for the dominant leg. Each session consisted of 4 sets of 8 repetitions. The AEL leg press exercise was performed with isotonic resistance generated by an electrically driven motor device at two different percentages of the concentric one-repetition maximum (1-RM): 30% of 1-RM for the concentric action and 105% of 1-RM for the eccentric one. Participants in the control did not perform any training program. Both dominant trained leg (TL) and ipsilateral non-trained arm (NTA) total lean tissue mass were assessed by DXA, 1-RM, maximal voluntary isometric contraction (MVIC), and unilateral muscle power at different intensities, were assessed before and after training.

### 2.2. Subjects

Sixty-nine undergraduate students of sports science volunteered for the present study (40 men: 20.1 ± 2.2 years, 76.1 ± 7.8 kg, 178.9 ± 5.7 cm; and 29 women: 20.4 ± 2.0 years, 60.2 ± 7.1 kg, 165.1 ± 5.6 cm). They were randomly placed into an experimental training group (*n* = 46; 27 men, 19 women) or into a control group without training (*n* = 23; 13 men, 10 women). Randomization was performed using an online application (https://www.randomizer.at, accessed on 25 April 2018), using the randomization list option. The total list of participants was randomized so that for every three participants, the third one was assigned to the control group. Participants were healthy and physically active individuals, engaged in 6–8 h of physical activity per week. They had a previous history of systematic weight training and no previous musculoskeletal disorders for the last 6 months before training. Before and after the intervention, the physical activity level (IPAQ-2018) was registered. Participants in the control group who increased their physical activity level or performed any form of resistance exercise training were eliminated from the study. In addition, all participants were asked to maintain their sleeping, eating, and hydration habits during the study. Participants were not allowed to take ergogenic aids or change their eating habits for the duration of the study. They were also informed about the purpose of the study and the risks that might arise during training before giving their written consent of participation. The Board and the Ethics Committee of the University of León accepted the protocols of the study (ETICA-ULE-009-2018). All the protocols were performed by the participants including, the pre and post-tests, the training program, and two familiarization sessions.

### 2.3. Protocol

On four occasions prior to start the training program, all participants attended our lab. On day 1, DXA was performed. On next day the unilateral leg press and unilateral elbow flexors 1-RM test were performed. A total 48–72 h later, MVIC and muscle power at three different intensities (40, 60, and 80% of 1-RM) were tested for both unilateral leg press and elbow flexors exercises. One week after intervention, the assessment protocol was replicated in the same order at the same time of day. Measurements were performed in both trained leg (TL) and ipsilateral non-trained arm (NTA). An individual and randomized limbs order was performed before testing for each participant and replicated at post-tests. Each day that participants attended to the laboratory, a warm-up of 5-min on an elliptical device, 25 reps of high knees, 25 reps of butt kicks, and 2 sets of 10 squat repetitions with their own body weight was performed.

### 2.4. Dual Energy X-ray Absorptiometry Analysis

DXA was performed using a Lunar Prodigy^®^ whole-body scan (GE Medical Systems, Madison, WI, USA). Manual analysis was performed to independently estimate total thigh and total arm lean mass (Encore^®^ 2009 software, Lunar Corp., Madison, WI, USA). Briefly, one rectangle mark was generated using the lower margin of the ischial tuberosities and the lower margin of the femoral condyles as thigh reference points. Lean mass was then calculated for the entire thigh [20]. Subsequently, a secondary rectangle mark was generated from the surgical neck of the humerus to the end of the fingers as arm reference points. Lean mass was then calculated for the entire arm [21]. Finally, lean tissue mass estimation in both total thigh and total arm slices created was calculated using Encore software for both TL and ipsilateral NTA.

### 2.5. Unilateral Maximal Dynamic Strength (1-RM)

The 1-RM test for the lower limb was performed using an inclined leg press device with an inclination of 45° (Gerva-Sport, Madrid, Spain). Participants performed one repetition from 90° to full extension (180°) with a load corresponding to approximately 3-RM. The start point (90° of knee flexion) was measured with a goniometer by an individual researcher and was limited by a security chain to mark the beginning of each repetition. When the participant succeeded in overcoming a given load, weight was increased by 10 kg; and weight was decreased by 5 kg when failed. The 1-RM test was concluded when the participant failed to overcome a given load in two successive attempts. The unilateral 1RM of leg press was obtained within 3 to 6 attempts and with a recovery time of 2 min in between. During testing, all participants were asked to place the non-trained leg with the knee flexed yet relaxed and the foot posed properly on the ground.

Unilateral elbow flexors 1-RM test was conducted on a Scott weight-stack device (Gerva-Sport, Madrid, Spain). Participants were positioned placing the chest and the axillary fossa in the Scott bench. Participants were asked to place the resting arm with the elbow extended and the hand propped on the ipsilateral knee. Start (0° of elbow flexion) and finishing points were measured with a goniometer by an individual researcher. In order to know that the finishing point was reached in each 1-RM test repetition, a linear encoder (T-FORCE Dynamic Measurement System, Ergotech Consulting S.L., Murcia, Spain) and the associated software (T-Force v. 2.28) was used to assess the distance covered during performance. Participants performed one repetition from full extension (0°) to 90° of elbow flexion with an approximate load of 3-RM. When the participant succeeded in overcoming the given load, the weight was increased by 2–4 kg, and the weight was decreased by 1–2 kg when failed. The test was concluded when the participant failed to overcome the given load in two successive attempts. The unilateral 1RM of elbow flexors was obtained within 3 to 6 attempts and with a recovery time of 2 min in between. The 1RM test for each limb occurred twice, the first one was 3–5 days before the training period, and the second was 3–5 days after the training period.

### 2.6. Unilateral Maximal Voluntary Isometric Contraction

Leg press MVIC was assessed at 90° of knee flexion in the same leg press device described above. The leg press device was equipped with a force transducer (GLOBUS Ge: S-Beam KM 1506 K, Art No 124 108, Megatron Elektronic AG, Putzbrunn, D, input voltage: ±5 V). The sensor was integrated into the security chain between the weights bar and the leg press seat, parallel to the 45° inclined middle rail. Similarly, elbow flexors MVIC was assessed at 90° of elbow flexion, employing the same Scott device and placing participants in the same position described above. In that case, the force sensor was integrated into a chain placed at 45° between the floor and the handgrip. Participants were instructed to perform two 5-s ramped maximal isometric contractions for each exercise tested. Test trials with verbal instructions to perform with maximum effort were given continuously. Data from the force transducer were sampled at 1000 Hz, and a 2-min recovery period between attempts was allowed. Only the best repetition performed with each limb was used for further analysis.

### 2.7. Unilateral Muscle Power at Different Intensities

In order to assess lower limb muscle power, three sets of three concentric-only repetitions each were executed unilaterally from 90° knee flexion to full extension (0°) on the above-mentioned leg press device. In the case of elbow flexion, the test started from a full elbow extension (0°) to 90° flexion, in the same device described above. The recovery time between sets for each test consisted of 2 min. Each attempt started from a complete static position to avoid the aid of the stretch-shortening cycle (SSC). Each set was performed on a percentage of the 1RM load (40%, 60%, and 80%, respectively), with an individual and randomized order, which was replicated at post-test. Participants were encouraged to execute the concentric contraction as fast as they possibly could. Mean power for each repetition was sampled at 1000 Hz using a linear encoder (T-FORCE Dynamic Measurement System, Ergotech Consulting S.L., Murcia, Spain) and the associated software (T-Force v. 2.28). The best repetition performed at each load was used for further analysis.

### 2.8. Training Program

Participants were randomly placed into an experimental training group or a control group without training. All participants included in the experimental group (men (*n* = 27); and women (*n* = 20)) completed 10-weeks (20 sessions) of an eccentrically accentuated unilateral leg press training program, using an electric motor device (Exentrix, SmartCoach™, Stockholm, Sweden) [20]. Volunteers trained 2 times per week with at least 48 h of rest between sessions [22]. Following a standardized cycling warm-up, participants performed 4 sets of 8 maximal unilateral (dominant leg) coupled concentric and eccentric muscle actions in a custom-made horizontal leg press device. The electric motor device was configured in isotonic mode (i.e., constant load during exercise) using the device’s software settings (Exentrix PC Interface-V2.4, SmartCoach™). Hence, two different intensities were employed: a submaximal load of 30% of 1-RM in the concentric action and a supramaximal load of 105% of 1-RM in the eccentric action. According to the manufacturer’s instructions, a single hoist with a simple mobile pulley was employed to duplicate the force generated by the motor. The transitional time between phases (eccentric/concentric) was established at minimum according to the system’s parameters. Participants were required to push with maximal effort through the entire concentric action, which ranged from 90° of knee flexion to nearly full extension. At the end of the concentric contraction, the motor strap rewinds back, starting the eccentric contraction. For each participant at each session and before initiating the first set, the range of motion (ROM) was set up from full extension (0°) to 90° knee flexion through a manual goniometer. Besides the fact that one of the characteristics of the electrically driven motor device is enabled to produce the eccentric-overload throughout the whole ROM [20,23]. Therefore, all training participants were instructed to stop the movement before reaching the end of ROM. They were also instructed to position the non-training leg extended on the floor and not to avail of it to produce force during the execution.

The resistance provided by the electric motor (a brushless DC motor) was controlled by a custom-designed power driver (SmartCoach™, Stockholm, Sweden) that controls in closed-loop both speed and torque variables. For the speed control loop, a high-precision incremental encoder is used to measure the actual speed. For the torque control loop, the current measured in the motor winding is used instead. With a technique used in automation and robotics, the actual torque (T, Nm) is computed from the current. The motor is coupled directly to a steel shaft (D = 25 mm) over which the rope is wound. Since the shaft is supported by a low-friction ball bearing, and the coupling between subject and motor is direct (no gears, no pulleys), the force can be computed by dividing the torque by the lever arm b = D/2 + d/2, where d is the rope diameter. Hence, F = T/b. Thus, mean and peak force and power were measured at each repetition for both concentric and eccentric contractions (SmartCoach™, Stockholm, Sweden), with feedback shown on the computer monitor in real-time. All participants of the training group had two familiarization sessions before training, while the non-training group (control) was instructed not to get involved in any strength training program during the whole study period.

### 2.9. Statistical Analysis

Statistical analyses were performed using SPSS v.26.0.0 (SPSS Inc., Chicago, IL, USA) and R version 4.1.9. Results are expressed as mean ± SD. Data distribution was examined for normality using the Shapiro–Wilk test. A four-way analysis of variance with repeated measures (group × time × limb × sex) and Bonferroni post hoc was used to investigate differences in variables after training within participants, between limbs, and between sex. The effect size (ES) was calculated for interactions between groups using Cohen’s guidelines. Threshold values for ES were >0.2 (small), >0.6 (large) and >2.0 (very large) [24]. In addition, Pearson’s r was used to determine linear correlations between the gains of the TL and the percentage of gain of the NTL, including all training groups. The differences observed between percentage changes after training in each leg were calculated by the paired samples t-test. The significance level was set to *p* < 0.05.

Additionally, the magnitude of changes of each limb for the experimental group was compared between men and women. With that purpose, the standardized of change mean differences (SCMD) were calculated using the following equations [25]:$$\mathrm{SCMD}=c\left({\mathrm{df}}_{\mathrm{men},\mathrm{women}}\right)\cdot \;\left[\frac{\left({\bar{X}}_{\mathrm{post},\mathrm{men}}-{\bar{X}}_{\mathrm{pre},\mathrm{men}}\;\right)-\left({\bar{X}}_{\mathrm{post},\mathrm{women}}-{\bar{X}}_{\mathrm{pre},\mathrm{women}}\right)}{{\bar{S}}_{\mathrm{pre}}}\right]$$
where df_men,women_ refers to the degree of freedom of men and women of the experimental group, respectively. Where S_pre_ is the pooled standard deviation of the experimental group in the pre-test, which was calculated by:$${\bar{S}}_{\mathrm{pre}}=\sqrt{\frac{\left(n_{\mathrm{men}}-1\right)\cdot {\;S}_{\;\mathrm{pre},\mathrm{men}}^{2}+\left(n_{\mathrm{women}}-1\right)\cdot S_{\;\mathrm{pre},\mathrm{women}}^{2}\;}{n_{\mathrm{men}}+{\;n}_{\mathrm{women}}-2}}$$
where S^2^_pre,men_ refers to the variance of pre-measurement in men group and S^2^_pre,women_ refers to the variance of pre-measurement in women group.

In addition, c(df_men,women_) is the correction factor, which was obtained by [26]:$$c\left({\mathrm{df}}_{\mathrm{men},\;\mathrm{women}}\right)=1-\left[\frac{3}{4\;\left(n_{\mathrm{men}}+{\;n}_{\mathrm{women}}-2\right)-1}\right]$$

The variance of the SCMD was computed as:$$\begin{array}{cc} S_{\;\;\;SCMD}^{2}= & {\left[c\left({\mathrm{df}}_{\mathrm{men},\mathrm{women}}\right)\right]}^{2}\cdot \;2\;\left(1-r\right)\cdot \left[\frac{n_{\mathrm{men}}+{\;n}_{\mathrm{women}}}{n_{\mathrm{men}}\cdot {\;n}_{\mathrm{women}}}\right]\cdot \left[\frac{n_{\mathrm{men}}+{\;n}_{\mathrm{women}}-2}{n_{\mathrm{men}}+{\;n}_{\mathrm{women}}-4}\right]\\ & \;\cdot \left[1+\frac{n_{\mathrm{men}}\cdot {\;n}_{\mathrm{women}}\cdot {\mathrm{SMD}}^{2}}{2\cdot \left(1-r\right)\cdot \left(n_{\mathrm{men}}+{\;n}_{\mathrm{women}}\right)}\right]-{\mathrm{SCMD}}^{2} \end{array}$$
where r is the average correlation coefficient between the pre and post measurements in both sexes. The correlation coefficient between pre-post measurements for both sexes was computed from the standard deviation of change score (SD_diff_) in each sex condition. The SCMD was considered trivial (<0.20), small (0.20–0.59), moderate (0.60–1.19), large (1.20–1.99), and very large (>2.00) [27]. Variance estimations between variables were calculated using a random-effects model (i.e., Hartung–Knapp/Sidik–Jakman adjustment (HKSJ)) with a 95% confidence interval (CI_95%_). The consistency of the effects found was assessed using the heterogeneity (I^2^) and Tau-square tests (τ^2^) tests, with I^2^ being considered small (<25%), moderate (25–49%), and high (>50%). In addition, τ^2^ and prediction interval (PI) were included because τ^2^ cannot readily point to the clinical implications of the unobserved heterogeneity [28]. The prediction interval allows a better clinical evaluation of the results obtained because it represents the range in which the effect size of a future study conducted on the topic will most likely be (i.e., probability of true-positive effect). Prediction intervals and the probability of the true-positive effects calculations were performed in accordance with previous studies [28].

## 3. Results

The average peak power for the concentric phase increased (*p* < 0.01) from the first to the last training session in both sexes, but the increase was similar between men (438.5 ± 72.3–791.1 ± 105.1 W) and women (349.7 ± 67.4–642.9 ± 120.5 W). The mean eccentric-overload in terms of power relative to the concentric average peak power was similar between groups, 67.7 ± 8.0% in men (CON: 521.8 ± 107.9 W; ECC: 875.8 ± 106.6 W), and 59.3 ± 36.1% in women (CON: 496.4 ± 163.0 W; ECC: 746.0 ± 140.9 W). The peak force generated in the eccentric phase relative to the concentric average peak force was 74.6 ± 1.1% in men (CON: 318.5 ± 9.9 N; ECC: 1256.5 ± 29.1 N), and 70.7 ± 3.3% in women (CON: 307.6 ± 12.0 N; ECC: 1053.3 ± 36.0 N). The mean concentric velocity (men: 2.03 ± 0.34–2.99 ± 0.43 m/s; women: 1.80 ± 0.31–2.71 ± 0.22 m/s) and the peak concentric phase velocity (men: 2.72 ± 0.40–3.68 ± 0.30 m/s; women: 2.30 ± 0.30–3.69 ± 0.21 m/s) were similar between sexes, but the mean eccentric phase velocity was significant slower (*p* < 0.01) in men (1.06 ± 0.21–1.42 ± 0.17 m/s) and women (0.82 ± 0.15–1.39 ± 0.28 m/s).

Table 1 and Table 2 show the results of the pre-post changes observed in the TL and in the ipsilateral NTA in men (Table 1) and women (Table 2) of both experimental and control groups. No significant correlations were observed between the magnitude of change in the TL and in the NTA for any variable (1-RM: r = 0.148, *p* = 0.333; MVIC: r = −0.259, *p* = 0.083; concentric mean power at low intensity: r = −0.128, *p* = 0.406; concentric mean power at medium intensity: r = −0.112, *p* = 0.476; concentric mean power at high intensity: r = −0.146, *p* = 0.368; TLM: r = −0.269, *p* = 0.078).

In addition, the comparison between sexes for the magnitude of change in all outcomes in the experimental group showed non-statistically significant differences (t-value = 1.69, *p* = 0.119) by 0.24 SCMD [−0.07, 0.55] in favor of men (Figure 1). Although the magnitude of changes showed in the TL was significant higher in favor of men participants (0.25 SCMD [0.12, 0.39], t-value = 4.72, *p* = 0.005), the NTA also showed non-statistically significant differences (0.23 SCMD [−0.57, 1.04], t-value = 0.74, *p* = 0.492) in favor of men participants (Figure 1).

## 4. Discussion

The present study examined the effects of unilateral accentuated eccentric loading RT on changes in muscle mass and function of the trained leg (TL) and the ipsilateral non-trained arm (NTA) in both men and women. After 20 sessions of the eccentric-overload training protocol, comparable increases in the maximum unilateral dynamic and isometric strength and unilateral muscle power at different loads were found for both TL and NTA (Table 1 and Table 2). These results, although non-significant correlations were found between the magnitude of the changes in any variable between TL and NTA, showed a strong lower-to-upper body cross-transfer effect of unilateral lower limb RT with eccentric-overload in strength-related outcomes. However, no significant gains in lean tissue mass were observed in the NTA after the intervention protocol. In addition, men showed general greater gains in the TL when the magnitude of the changes of all variables was compared (Figure 1), but this was not observed in the NTA, showing similar changes between sexes.

It has been shown that a short-term RT with eccentric-overload induced by either flywheel and electric motor devices effectively increased muscle strength, power, lean mass, and vertical jump performance in the TL [20,29]. These results are consistent with the results showed by the TL in the present study. Although, the underpinning mechanisms of the robust neuromuscular enhancements shown, yet have not been broadly elucidated, it may be explained by task-specific differences and load used for both concentric and eccentric contractions [30]. Since higher loading of the eccentric phase may lead to an enhanced motor cortex activation [31], compensating spinal inhibition and enhancing neural drive [32], incorporating selective recruitment of high threshold motor units and stimulating Type Ia afferent nerves [33], which in turn results in decreasing motor-evoked potentials and H-reflex response and a specialized motor unit activation pattern during lengthening compared with shortening [34,35]. Hence, these task-specific neural changes may optimize performance on those activities in which the stretch-shortening cycle is involved [30]. In addition, muscle hypertrophy is also a likely contributor to the favorable performance changes observed after AEL [36]. Thus, AEL has proven to favor increases in fascicle length and hypertrophy of the distal portions of a muscle (traditionally associated with eccentric-only training) while maintaining the proximal muscle changes (i.e., pennation angle increases and greater hypertrophy mid-muscle) associated with traditional RT [37]. Therefore, changes in explosive performance may be attributed to increased contraction speed via in-series specific hypertrophy from the eccentrically overloaded stimulus, while changes in strength may be due to in-parallel specific hypertrophy from the concentric stimulus [37]. Despite architectural changes, AEL has been shown to lead to enhancements in factors involved in anabolic signaling, such as several insulin-like growth factors, myogenic regulatory factors, or testosterone [38], which in turn results in Type IIx and IIa fiber-specific cross-sectional area gains [38], and Type I fiber-type percentage decrease while Type IIx and IIa fiber-type percentage increase in those muscle groups involved in an accentuated eccentric training program [39]. All these factors taken together may explain the relatively large increases observed in 1-RM, MVIC, mechanical power at different intensities, and muscle mass in the TL after an RT program with eccentric-overload. In addition, it might explain the superiority of AEL training-induced effects in men compared to women for the TL.

However, the effects of the unilateral AEL on the non-trained contralateral limb have not been explored. Nonetheless, it is well known that unilateral resistance exercise increases muscle strength of the contralateral non-trained homologous muscles, which is referred to as the cross-education or cross-transfer effect [1,2,3,15,40]. Green and Gabriel [40], in their review paper, reported that the magnitude of the increase in muscle strength for the contralateral non-trained limb ranged from 52 to 80% of that of the trained limb. In fact, it has led us to hypothesize that not only the non-trained homologous limb can show cross-transfer changes, but also the untrained ipsilateral upper limb. In the present study, an increase in 1-RM and MVIC in the NTA elbow flexors was observed after a 10-week AEL program in both men (1-RM: 10.5%; MVIC: 14.7%) and women (1-RM: 20.6%; MVIC: 69.4%). Although women showed larger effect sizes in the changes in NTA for concentric and isometric strength, no significant differences were found for the magnitude of the change between sexes. In addition, these results are in line with previous studies. Similarly, Ben Othman and colleagues [19] reported significant gains (3.3%) in the elbow flexion unilateral MVIC. Unfortunately, Ben Othman et al. [19] reported no data regarding the elbow flexors 1-RM to compare with our study findings. They referred to the lower-to-upper body cross-transfer effect from ipsilateral TL to ipsilateral NTA as the global training effects. Since, as has been demonstrated in our study, the magnitude of the ipsilateral NTA strength gains is not dependent on those obtained ipsilaterally in the TL.

In addition, the present study results suggest that with the implementation of lower limb AEL, the force-velocity relationship might be better expressed not only in the ipsilateral TL but also in the ipsilateral NTA. Since muscle power at low (men: 59.0%, women: 72.6%), medium (men: 47.1%, women: 60.8%) and high loads (men: 19.6%, women: 53.3%) significantly increased in the NTA, with difference between sexes favors to men at low and medium loads (Figure 1). However, these results should be considered with caution since this study was the first one to analyze and compare changes in concentric mean power at different intensities of the 1-RM in the elbow flexion exercise after a lower limb RT intervention. Moreover, no significant correlation was found between the magnitude of changes in the TL and in the NTA. Nonetheless, as proposed down below, neural adaptations are likely responsible for strength and strength-related changes observed in the NTA.

Finally, even though both men and women showed similar gains in total thigh lean mass (4.1 and 3.3%, respectively) in the TL after the RT program. However, the NTA did not show any significant gains on lean mass. Hence, as previous studies have proposed regarding the cross-education effect [1,3], the early contralateral increases in muscle force are not associated with lean mass gains in NTA. Which in turn, it is largely due to the fact that no significant vascular adaptations have been found [41], nor were any histological changes in hypertrophy levels, in enzyme concentration, in contractile protein composition alteration, in fiber type, or in cross-sectional area (CSA) in the ipsilateral NTA after lower limb unilateral RT [5].

Therefore, the similar magnitude of increases in muscle function between TL and NTA found in the present study may be attributed only to neural adaptations [1,3,42]. However, the exact nature of the neural adaptations remains unclear. In trying to explain these adaptations, two theories are currently postulated that, although compatible with each other, may explain how the neural adaptation mechanisms occur [43]: (a) the “cross-activation” model, which suggests that adaptations to unilateral exercise extend to the opposite half of the body, and (b) the “bilateral access” model, which maintains that the motor schema of a unilateral task is accessible by trying to reproduce the same task in the opposite half of the body. Regarding the changes observed in the nervous system, these can occur at peripheral, medullar, subcortical, and cortical levels [44]. At peripheral and medullar levels, various studies suggest the existence of alterations in the synchronization of motor units and of neural conductivity similar to those observed in the trained side [5]. At subcortical and cortical levels, there is some evidence confirming the existence of neural interaction between the two hemispheres [5,45]; thus, supporting the cross-activation model suggested by Ruddy and Carson (2013). In addition to all the above mentioned, it seems that motor learning provokes cortical reorganizations [5] and that unilateral exercise produces inter-hemispheric plasticity [46], thus supporting the bilateral access model. More recent research employing transcranial magnetic stimulation has reported increases in corticospinal excitability of the contralateral biceps brachii [13], as well as decreased corticospinal inhibition of the contralateral quadriceps [47] when lower limb resistance exercise was performed. Kidgell et al. [13] reported that eccentric-only training reduced intracortical inhibition, silent period duration, and increased corticospinal excitability greater when compared with concentric-only training. In addition to possible neural adaptations, novice, untrained and non-experienced subjects might show greater adaptations due to greater coordination and learning effects, whereas well-trained individuals may emphasize more neural (recruitment and rate coding) and morphological adaptations [48]. It should be noted that motor control, learning, coordination, and some structural changes were enhanced to a greater degree in the trained leg, whereas the heterologous muscle NTA adaptations would have been more reliant upon neural adaptations [13,44,45,47,49].

There were some limitations in the present study. Although it is interesting to see relatively large increases in muscle function in the present study, it should be investigated further for muscle adaptations also in the heterologous non-trained limb (NTL) in order to compare the magnitude of changes in the NTA and NTL. Furthermore, despite the fact that the sample was very homogeneous in terms of age and training experience, it would be interesting to replicate this study with different age groups in order to study the effect of age and health status. Moreover, although the involvement of the NTA in the exercise was minimal (hands were placed in the handholds of the device seat), we did not assess the muscle activation (i.e., electromyography) during exercise in the elbow flexors. This is a limitation of the present study since it could provide additional details about the changes in the NTA. In fact, the measurement of muscle activation during exercise and corticospinal excitability and inhibition could provide deeper insights into the lower-to-upper body cross-transfer effects. Moreover, it is warranted to compare the magnitude of the lower-to-upper body cross-transfer effect between traditional concentric-eccentric, eccentric-only, and accentuated eccentric loading. Finally, it appears that unilateral accentuated eccentric loading is an interesting asset in clinical and rehabilitation settings, specifically with patients who have an immobilized limb or neurological disorders (such as multiple sclerosis or stroke patients), allowing them to maintain their strength levels and muscle mass in the ipsilateral immobilized or affected limb. However, the feasibility of this approach in a clinical environment remains unknown.

## 5. Conclusions

In conclusion, the present study demonstrated that 10 weeks (20 sessions) of RT with eccentric-overload led to significant gains in concentric 1-RM, MVIC, and muscle power at different loads in both TL and ipsilateral NTA in a similar magnitude for physically active young men and women. Additionally, the training group showed significant increases not only in muscle function but also in total lean mass in the TL, whereas the total NTA lean mass did not change. Since no neurophysiological measurement was performed in the present study, more studies are warranted to investigate the mechanisms underpinning the large lower-to-upper body cross-transfer effects induced by AEL. It is also necessary to investigate the effectiveness of this training approach in clinical and rehabilitation settings.

## Figures and Tables

**Figure 1 medicina-57-00445-f001:**
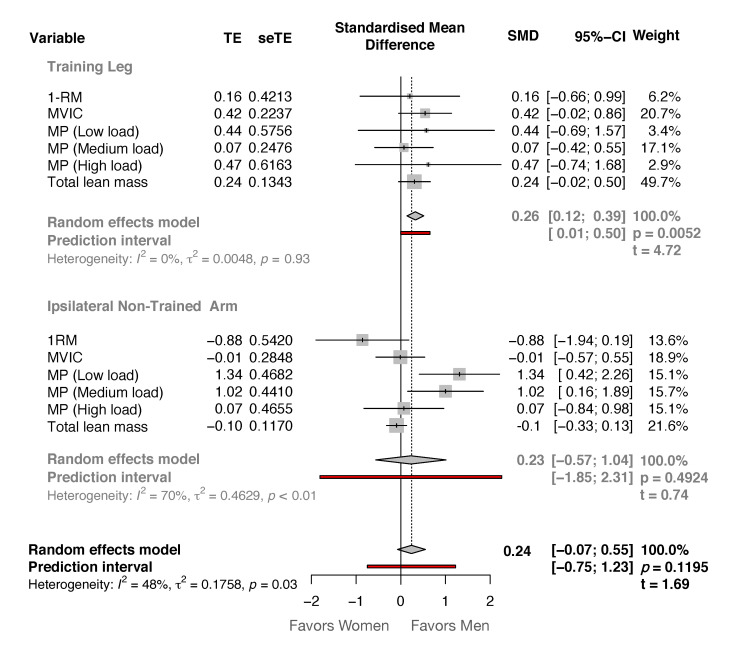
Forest plot showing the difference in the standardized mean differences between men (right side of the vertical black line) and women (left side of the vertical black line) in the training leg and in the non-trained arm.

**Table 1 medicina-57-00445-t001:** Changes (mean ± SD) in unilateral one-repetition maximal strength (1-RM), unilateral maximal voluntary isometric contraction (MVIC), concentric mean power at low (PLL, 40% of 1-RM), medium (PML, 60% of 1-RM), and high loads (PHL, 80% of 1-RM), and total lean tissue mass (TLTM), for the trained leg (TL) and non-trained arm (NTA) for the training group and control group before (Pre) and after training (Post) in men. The *p*-value for the comparison between pre- and post-training values by Bonferroni test, and effect size (ES) for the changes, the magnitude of change (%) are shown for each limb.

Men	Trained Leg					Ipsilateral Non-Trained Arm				
	Pre	Post	*P*	ES	%	Pre	Post	*P*	ES	%
**Training group** (*n* = 27)										
1-RM (kg)	183.3 ± 8.5	202.6 ± 5.5	0.013	0.72	10.5	32.9 ± 1.4	47.8 ± 1.5 ^^^	<0.001	2.02	45.2 *
MVIC (kg)	78.3 ± 3.6	89.8 ± 3.9	<0.001	0.67	14.7	19.6 ± 0.8	23.2 ± 0.7	<0.001	0.91	18.2
PLL (W)	426.3 ± 12.9	494.4 ± 12.8	<0.001	1.20	16.0	31.3 ± 2.1	49.7 ± 1.9 ^^^	<0.001	1.71	59.0 *
PML (W)	448.9 ± 15.7	496.5 ± 13.1	<0.001	0.78	10.6	52.0 ± 2.9	76.5 ± 2.7 ^^^	<0.001	1.49	47.1 *
PHL (W)	417.1 ± 16.6	456.9 ± 15.4	0.033	0.46	9.5	57.1 ± 3.3	68.3 ± 3.2	0.002	0.68	19.6
TLTM (mm)	6528.9 ± 123.1	6797.6 ± 123.0	<0.001	0.45	4.1 *	3882.1 ± 78.4	3857.9 ± 79.0	0.371	0.06	−0.6
**Control group** (*n* = 13)										
1-RM (kg)	223.8 ± 12.3 ^^^	188.5 ± 7.9	0.002	0.99	−15.8	38.9 ± 2.0 ^^^	39.5 ± 2.1	0.710	0.06	1.6 *
MVIC (kg)	83.2 ± 5.4	87.4 ± 5.8	0.370	0.20	5.0	21.1 ± 1.3	23.5 ± 1.2	0.053	0.39	9.8
PLL (W)	520.7 ± 18.2 ^^^	491.1 ± 18.1	0.007	0.30	−5.7	41.8 ± 3.0 ^^^	40.2 ± 2.8	0.426	0.11	−3.8
PML (W)	545.3 ± 22.6 ^^^	588.9 ± 92.4	0.155	0.17	−3.7	67.9 ± 4.2 ^^^	64.2 ± 3.8	0.173	0.19	−5.5
PHL (W)	484.3 ± 24.0 ^^^	506.4 ± 22.1	0.404	0.18	4.5 *	77.4 ± 4.8 ^^^	57.7 ± 4.6	<0.001	0.77	−25.5
TLTM (mm)	7236.6 ± 177.4 ^^^	7163.4 ± 177.3	0.120	0.08	−1.0	4151.1 ± 113.1	4176.1 ± 113.9 ^^^	0.518	0.04	0.6

*: a significant (*p* < 0.05) difference for the magnitude of changes between TL and NTA. ^^^: a significant (*p* < 0.05) difference between groups.

**Table 2 medicina-57-00445-t002:** Changes (mean ± SD) in unilateral one-repetition maximal strength (1-RM), unilateral maximal voluntary isometric contraction (MVIC), concentric mean power at low (PLL, 40% of 1-RM), medium (PML, 60% of 1-RM) and high loads (PHL, 80% of 1-RM), and total lean tissue mass (TLTM), for the trained leg (TL) and non-trained arm (NTA) for the training group and control group before (Pre) and after training (Post) in women. The *p*-value for the comparison between pre- and post-training values by Bonferroni test, and effect size (ES) for the changes, the magnitude of change (%) are shown for each limb.

Women	Trained Leg					Ipsilateral Non-Trained Arm				
	Pre	Post	*P*	ES	%	Pre	Post	*P*	ES	%
**Training group** (*n* = 19)										
1-RM (kg)	113.9 ± 10.1	137.4 ± 6.6	0.015	0.86	20.6	12.9 ± 1.6 ^^^	21.8 ± 1.8 ^^^	<0.001	1.24	69.4 *
MVIC (kg)	52.8 ± 4.3	55.9 ± 4.6 ^^^	0.411	0.14	5.7	10.7 ± 1.0	14.3 ± 0.9 ^^^	<0.001	1.02	32.8 *
PLL (W)	251.7 ± 15.4	294.8 ± 15.4	<0.001	0.77	17.1	9.2 ± 2.5	15.7 ± 2.3	<0.001	1.04	72.6 *
PML (W)	279.1 ± 20.3	322.4 ± 17.0	0.001	0.69	15.5	16.5 ± 3.8	26.5 ± 3.4 ^^^	<0.001	1.18	60.8 *
PHL (W)	272.2 ± 24.0	275.2 ± 22.1	0.909	0.08	1.1	19.2 ± 4.8	29.4 ± 4.6 ^^^	0.039	0.99	53.3 *
TLTM (mm)	4268.5 ± 155.1	4407.6 ± 155.0 ^^^	0.001	0.31	3.3 *	2180.3 ± 98.9	2193.3 ± 99.6	0.703	0.04	0.6
**Control group** (*n* = 10)										
1-RM (kg)	142.5 ± 14.0	109.0 ± 9.1	0.010	0.49	−23.5	8.5 ± 2.3	8.6 ± 2.4	0.985	0.01	0.4
MVIC (kg)	39.2 ± 5.9	44.3 ± 6.4	0.311	0.29	13.1	10.3 ± 1.4	10.7 ± 1.2	0.781	0.12	3.5
PLL (W)	241.6 ± 29.3	240.6 ± 29.2	0.953	0.05	−0.4	7.3 ± 4.8	8.7 ± 4.5	0.662	0.77	19.4 *
PML (W)	237.4 ± 25.7	248.0 ± 21.5	0.504	0.21	4.5	10.2 ± 4.8	10.9 ± 4.4	0.816	0.15	7.1
PHL (W)	213.1 ± 28.8	222.8 ± 26.6	0.760	0.25	4.5 *	13.9 ± 5.8	14.3 ± 5.6	0.952	0.05	2.5
TLTM (mm)	3838.5 ± 202.9	3807.1 ± 202.1	0.555	0.05	−0.8	1904.9 ± 128.9	1895.1 ± 129.8	0.827	0.04	−0.5

*: a significant (*p* < 0.05) difference for the magnitude of changes between TL and NTA. ^^^: a significant (*p* < 0.05) difference between groups.

## Data Availability

The data presented in this study are available on request from the corresponding author. The data are not publicly available due to participants’ privacy.

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
