# Peer review of "Ipsilateral Lower-to-Upper Limb Cross-Transfer Effect on Muscle Strength, Mechanical Power, and Lean Tissue Mass after Accentuated Eccentric Loading"

_medicina, 2021, doi:10.3390/medicina57050445_

Round 1

Reviewer 1 Report

Dear authors,

it was a pleasure to read your well written original article on the ipsilateral lower-to-upper limb cross-transfer effect on muscle strength, mechanical power and lean tissue mass after accentuated eccentric load. In the introduction you describe different theories that may explain the “cross education effect” of unilateral resistance training. Your main hypothesis is formulated clearly, and the novelty of your study is emphasized. The methods are well suited to test your hypothesis and are described very detailed. Especially the statistical methods have been profoundly chosen. The results are presented soundly and illustrated well. Although there were no statistically significant correlations between the magnitude of the changes in any variable between TL and NTA, the current study clearly showed a strong lower-to-upper body cross transfer effect of the unilateral lower limb RT with eccentric-over-load in strength-related outcomes as the main finding. In the discussion you explain the effects of AEL, as enhanced motor cortex activation, muscle hypertrophy and enhancements in factors involved in anabolic signaling as possible reasons for the effects on the ipsilateral NTA. Up-to-date literature is used to underpin your hypothesis. The conclusion is supported by the study design and results.

Overall, you present a scientifically sound study with above average novelty and significance of the content. Only minor revision is warranted. Please see the suggestions below:

Abstract:

  • Please clearly state the study design: Prospective, randomized.
  • No changes in any variable were found for the control group.” (line 21). That´s not true, as there was a significant decrease in PHL of NTA in the male control group. Please rephrase.

Introduction:

  • Please write out the abbreviation AEL when using it the first time in the main text (line 75).

Methods:

  • Please explain the process of randomization more detailed. Please explain the rationale of assigning 2/3 of participants to the training group and 1/3 to the control group.
  • The training group underwent a standardized cycling warm up prior training on the leg press device. Did the control group also do the cycling exercise to safely rule out effects of the warm-up? Please state or discuss.
  • How did you assess compliance of the participants? Did you monitor the daily exercise activity of the control group? Was any medication or nutrinional supplements taken on a daily base?

Results:

  • Interestingly you report a significant decrease in PHL in the ipsilateral NTA in the control group of male participants by -25.5%. Is there any potential explanation for this finding?
  • Spelling mistake in line 322: in favor “if” man -> of

Author Response

Dear Reviewer 1,

The authors appreciate the positive and constructive comments provided regarding the manuscript (ID:medicina-1184983) entitled Ipsilateral lower-to-upper limb cross-transfer effect on muscle strength, mechanical power and lean tissue mass after accentuated eccentric loading. Please see our responses to your concerns below and the attached revised manuscript. We believe the manuscript has been significantly improved and now hopefully qualifies for acceptance in its current form.

  1. Abstract:
  • Please clearly state the study design: Prospective, randomized.

Author’s response: In the light of reviewer’s suggestion, information about study design is now added to the revised version of the manuscript.

  • No changes in any variable were found for the control group.” (line 21). That´s not true, as there was a significant decrease in PHL of NTA in the male control group. Please rephrase.

Author’s response: Due to the word limit in the abstract section established by the journal, we have limited ourselves to indicate that “no gains in any variable were found for the control group”.

  1. Introduction:
  • Please write out the abbreviation AEL when using it the first time in the main text (line 75).

Author’s response: Amended.

  1. Methods:
  • Please explain the process of randomization more detailed. Please explain the rationale of assigning 2/3 of participants to the training group and 1/3 to the control group.

Author’s response: We appreciate this comment. Randomization was performed using the online application: https://www.randomizer.at, using the randomization list option. The total list of participants was randomized, so that for every three participants, the third one was assigned to the control group. This information has been uploaded in the revised version of the manuscript. Since the main objective of the project was to compare the training-induced changes by AEL in both TL and NTA in men and women, we decided to use a larger sample size in the experimental group. To avoid possible statistical biases and errors derived from a lower statistical power as a consequence of heterogeneity in the sample sizes between groups, the variance of both groups was found to be similar, and A four-way analysis of variance with repeated measures (group x time x limb x sex) and Bonferroni post hoc was used.

  • The training group underwent a standardized cycling warm up prior training on the leg press device. Did the control group also do the cycling exercise to safely rule out effects of the warm-up? Please state or discuss.

Author’s response: Thank you for your question. The control group did not perform any resistance exercise training throughout the intervention, as indicated in the methods section. Likewise, they did not perform the warm-up established for the training sessions of the experimental group. In our humble opinion, conditioning activities used (i.e., 5-min cycling and dynamic stretching), due to its cyclical, brief and low intensity characteristics were not a sufficient stimulus to generate changes in dependent variables (strength, mechanical power and lean mass).

  • How did you assess compliance of the participants? Did you monitor the daily exercise activity of the control group? Was any medication or nutrinional supplements taken on a daily base?

Author’s response: In the case of experimental participant, each visit to the laboratory was recorded and compliance with the intervention was controlled using the training load control tool (SmartCoach™, Stockholm, Sweden). In the other hand, the control group participants were request to not perform any programmed resistance exercise during the study duration. This was controlled through a physical activity questionnaire (IPAQ-2018) before and after intervention. Those participants who increased their physical activity level or performed any form of exercises training were eliminated from the study. In addition, all participants were asked to maintain their sleeping, eating and hydration habits during the study. Participants were not allowed to take ergogenic aids or change eating habits for the duration of the study. We have included this information in the subjects section. Please see the methods section in the revised version of the manuscript.

  1. Results:
  • Interestingly you report a significant decrease in PHL in the ipsilateral NTA in the control group of male participants by -25.5%. Is there any potential explanation for this finding?

Author’s response: This is a very interesting question. Even though unloading-induced effects in control group were out of the scope of this study, previous research in which detraining effects were analyzed have attributed these detrimental effects to significant neuromuscular and muscle mass losses and muscle architecture changes (Blazevich, 2006). In this study we observed significant decreases on concentric mean power at 80% of 1-RM in NTA, but not in MVIC or 1RM. Similarly, previous studies have shown that muscular power decline faster in comparison with force production (Mujika & Padilla, 2001). Moreover, a 10-week period of inactivity may imply other neural changes that would explain this finding, such as a decrease in neural drive, and the consequent decrease in electromyographic activity during a maximal contraction, in addition to a decrease in the cross-sectional area of type IIX fibers (Hortobágyi et al., 1993).

  • Spelling mistake in line 322: in favor “if” man -> of

Author’s response: Amended.

Reviewer 2 Report

Strengths

To investigate the effects of unilateral accentuated eccentric loading on changes in lean mass and function of leg trained and ipsilateral non-trained arm in young men and women.

INTRODUCTION

Restructure the last paragraph because it leads to confusion.

Reflect the importance of work. What is the reason for this investigation? What application does it have outside the medical field?

MATERIALS AND METHODS

General design

The part that talks about the sample would be passed to the next subsection (“subjects”).

STATISTICAL ANALYSIS

The box with the formlas makes reading a bit difficult.

In line 260, Spre should appear as text, not in formula format.

CONCLUSIONS

Highlight the importance of the work.

Author Response

Dear Reviewer 1,

The authors appreciate the positive and constructive comments provided regarding the manuscript (ID:medicina-1184983) entitled Ipsilateral lower-to-upper limb cross-transfer effect on muscle strength, mechanical power and lean tissue mass after accentuated eccentric loading. Please see our responses to your concerns below and the attached revised manuscript. We believe the manuscript has been significantly improved and now hopefully qualifies for acceptance in its current form.

  1. INTRODUCTION
  • Restructure the last paragraph because it leads to confusion.

Author’s response: The proposed change has been made. Please see Introduction section.

  • Reflect the importance of work. What is the reason for this investigation? What application does it have outside the medical field?

Author’s response: Based on the lack of research analyzing the lower-to-upper body cross-limb effect, we designed a study to assess and compare training induced-effects not only on trained leg but also on non-trained arm. With the hypothesis that AEL performed by the lower limb may leads to significant gains in neuromuscular variables, which would have an important impact in clinical and rehabilitation settings, specifically with patients who have an immobilized limb or neurological disorders such as multiple sclerosis or stroke patients), allowing them to maintain their strength levels and muscle mass in the ipsilateral immobilized or affected limb.

According with reviewer’s suggestions the last introduction paragraph has been updated including this information. Please see Introduction section in the revised version of the manuscript.

  1. MATERIALS AND METHODS
  • General design: The part that talks about the sample would be passed to the next subsection (“subjects”).

Author’s response: Amended.

  1. STATISTICAL ANALYSIS
  • The box with the formulas makes reading a bit difficult.

Author’s response: Due to the journal’s template we are not able to delete formula’s boxes. 

  • In line 260, Spre should appear as text, not in formula format.

Author’s response: Amended.

  1. CONCLUSIONS
  • Highlight the importance of the work.

Round 2

Reviewer 2 Report

Thank you for the effort made in answering each comment point by point.